# Chitosan–Imidazolium Core–Shell Nanoparticles of Gd-Mn-Mo Polyoxometalate as Novel Potential MRI Nano-Agent for Breast Cancer Detection

**DOI:** 10.3390/mi14040741

**Published:** 2023-03-27

**Authors:** Fahimeh Aminolroayaei, Daryoush Shahbazi-Gahrouei, Mahboubeh Rostami, Seyed Hossein Hejazi, Amin Farzadniya

**Affiliations:** 1Department of Medical Physics, School of Medicine, Isfahan University of Medical Sciences, Isfahan 81746-73461, Iran; 2Novel Drug Delivery Systems Research Centre and Department of Medicinal Chemistry, School of Pharmacy and Pharmaceutical Sciences, Isfahan University of Medical Sciences, Isfahan 81746-73461, Iran; 3Department of Parasitology and Mycology, School of Medicine, Isfahan University of Medical Sciences, Isfahan 81746-73461, Iran; 4Department of Radiology, Askarieh Hospital, Isfahan 81997-53966, Iran

**Keywords:** nanoparticles, gadolinium–manganese–molybdenum polyoxometalate (POM), chitosan–imidazolium (CSIm), breast cancer, magnetic resonance imaging

## Abstract

Polyoxometalates (POMs) are mineral nanoclusters with many advantages in various diagnostic fields, in particular cancer detection. This study aimed to synthesize and evaluate the performance of gadolinium–manganese–molybdenum polyoxometalate (Gd-Mn-Mo; POM) nanoparticles coated with chitosan–imidazolium (POM@CSIm NPs) for detecting 4T1 breast cancer cells by magnetic resonance imaging in vitro and in vivo. The POM@Cs-Im NPs were fabricated and characterized by FTIR, ICP-OES, CHNS, UV–visible, XRD, VSM, DLS, Zeta potential, and SEM. Cytotoxicity, cellular uptake, and MR imaging in vivo and in vitro of L929 and 4T1 cells were also assessed. The efficacy of nanoclusters was demonstrated using MR images of BALB/C mice bearing a 4T1 tumor in vivo. The evaluation of the in vitro cytotoxicity of the designed NPs showed their high biocompatibility. In fluorescence imaging and flow cytometry, NPs had a higher uptake rate by 4T1 than L929 (*p* < 0.05). Furthermore, NPs significantly increased the signal strength of MR images, and its relaxivity (r1) was calculated as 4.71 mM^−1^ s^−1^. MR imaging also confirmed the attachment of nanoclusters to cancer cells and their selective accumulation in the tumor region. Overall, the results showed that fabricated POM@CSIm NPs have considerable potential as an MR imaging nano-agent for early 4T1 cancer detection.

## 1. Introduction

According to the World Health Organization (WHO), breast cancer accounts for 10% of all cancers diagnosed worldwide each year. This type of cancer is the second leading cause of cancer death in women after lung cancer [1]. Of course, early diagnosis and advanced treatment can reduce mortality in women with breast cancer [2,3]. Various imaging modalities, such as mammography, sonography, computerized tomography (CT) scan, and magnetic resonance imaging (MRI), can be used for diagnosis. Among these modalities, MRI is one of the best clinical imaging techniques due to some unique features, such as its non-invasive nature, lack of ionizing radiation, high spatial resolution, high soft-tissue contrast, and no limitation in penetration depth. Although MRI is the best technique for soft tissue imaging, the prolonged relaxation time of water protons leads to poor tissue differentiation; as a result, the soft tissue has difficulty being diagnosed as malignant [4]. Therefore, MRI sensitivity is low, but fortunately, magnetic resonance contrast agents such as gadolinium, iron oxide, and manganese can reduce the relaxation time of water protons [5]. As a result, they improve contrast and increase the sensitivity of magnetic resonance detection [6]. With advances in nanomedicine, integrating nanotechnology and molecular imaging that uses nanoparticles as a contrast agent has received particular attention in treatment and diagnosis [7].

Polyoxometalates (POMs) are nanosized poly union clusters of high-capacity transition metal ions (tungsten, molybdenum, vanadium, and niobium) and oxygen atoms with structural diversity attracted much interest from researchers in the cancer area. The paramagnetic properties, besides their anticancer activities, make them a promising class of contrast nano-agents in cancer diagnosis [8,9,10]. The association of POMs in molecular magnetism is based on the ability of these metal oxide clusters to act as chelating ligands with a large number of magnetic centers at specific locations in their molecular structures [11].

In 2021, Zhang S et al. [12] conducted a study on the potential of a supermolecular complex for MRI/CT imaging and photothermal therapy. This study prepared K-Gd solution by stirring the Lys-Asp-His-Cys-His-Val-Thr-Pro peptide Tyr with a GdW10 aqueous solution. Then the K-Gd solution was irradiated under UV light for photoreduction to obtain a reduced (rK-Gd) solution. The POM polyanion was covered with cationic organic molecules to overcome the binding of bare POM to biomolecules and their sensitivity to physiological conditions. They showed that this cluster has T_1_-weighted MR imaging (relaxivity or r_1_ K-Gd of 11.13 mM^−1^ s^−1^), possesses a high Hounsfield unit value for enhanced CT, and exhibits stable and outstanding photothermal behavior and enhanced antibacterial activity. In another study, Luyan Zong et al. synthesized GdW_10_@Ti_3_C_2_ in 2018 [13]. Their result showed the photothermal conversion capability of Ti_3_C_2_ MXene for tumor hyperthermia, as well as the capability of Gd in POM for MRI and W for CT. In this study, r_1_ of GdW_10_@Ti_3_C_2_ was calculated to be 7.09 mM^−1^ s^−1^. The in vitro function of this structure as an MRI contrast agent showed a positive concentration-dependent enhanced effect on the images. The in vivo function showed a gradual accumulation of structure in tumor tissues through the effect of EPR (enhanced permeability and retention).

In a study by Rakesh Gupta, [14] two pentanuclear Mn^II^-substituted sandwich-type polyoxotungstate complexes, [{Mn(bpy)}_2_Na(H_2_O)_2_(MnCl)_2_{Mn(H_2_O)}(AsW_9_O_33_)2]^9−^ and [{Mn(bpy)}_2_Na(H_2_O)_2_(MnCl){Mn(H_2_O)}_2_(SbW_9_O_33_)_2_]^8−^, were reported for the first time. According to their study, temperature-dependent direct-current magnetic susceptibility data indicated ferromagnetic interactions between Mn ions, and alternating-current magnetic susceptibility measurements with a direct-current biased magnetic field showed the existence of a ferromagnetic order for both samples.

Li et al. [15] synthesized K6MnSiW11O39 and K8MnP2W17O61 and evaluated their potential for MRI contrast agent (r_1_ of MnSiW_11_ and MnP_2_W_17_ was 12.1 mM^−1^ s^−1^ and 4.7 mM^−1^ s^−1^, respectively). Their result showed that the T1-relaxivities of two structures were higher or similar than gadolinium–diethylenetriamine penta-acetic acid (Gd DTPA), and the relaxivities, r1, of the two MnPOMs were higher than that of manganese (II)–dipyridoxal diphosphate (MnDPDP).

However, the high and long-term toxicity of inorganic POMs prevents their clinical use; in addition, the dissociation of naked POMs during metabolism is inevitable. Therefore, the encapsulation of POMs as negatively charged metalate anions, with mostly cationic structures, is necessary to reduce their inherent toxicity and obtain better efficiency via improving their structural stability [8].

One of the cationic polymeric structures for POMs coating is chitosan, a naturally occurring deacetylated product of chitin that has intrinsic properties for biomedical applications [16]. The benefits of chitosan include increased physicochemical stability, controlled release, improved tissue/cell interaction, increased bioavailability of drugs or active substances, and drug efficacy [17], as well as its low cytotoxicity, low cost [18], and low immunogenicity [19]. To enhance the electrostatic interactions and improve the coating efficiency, inducing more positive moieties on the chitosan structure could be a reasonable strategy.

Imidazolium-modified chitosan (CSIm) is one of those polymeric scaffolds used before as a potential carrier for POMs [20]. Therefore, coating polyoxometalates with CSIm leads to a higher coating efficiency than chitosan alone. That is the reason why this study aimed to produce the POM@CSIm NPs and evaluate their potential as an MR imaging nano-agent in in vitro and in vivo study.

## 2. Experimental

### 2.1. Materials and Instruments

Ammonium heptamolybdate ((NH_4_)_6_Mo_7_O_24_), manganese acetate (Mn (CH3COO)2·4H2O), glacial acetic acid (GAA, CH_3_COOH), isonicotinic acid (Hina: C_6_H_5_NO_2_ and Ina: C_6_H_4_NO_2_), N, N dimethylformamide (DMF, C_3_H_7_NO), gadolinium nitrate (Gd(NO_3_)_3_), and chitosan (CS) (average molecular weight 250 kDa, about 75–80% de-acetylated) were purchased from Sigma-Aldrich, Darmstadt, Germany. The 1-(2-chloroethyl)-3-methylimidazolium chloride was prepared based on an earlier procedure [21]. Hydrogen peroxide (H2O2), N-methylimidazole, 1,2-dichloromethane, acetone, and ethanol (C_2_H_5_OH) all were purchased from local vendors. Antibiotic solutions (penicillin–streptomycin), 3-(4, 5-dimethylthiazol-2-yl)-2,5diphenyltetrazolium bromide (MTT), and dimethyl sulfoxide (DMSO) were obtained from Sigma-Aldrich (Germany). Phosphate-buffered saline (PBS), trypsin, and fetal bovine serum (FBS) were purchased from Sigma-Aldrich (St. Louis, MO, USA). Roswell park memorial institute-1640 (RPMI) was obtained from a local vendor. The 4T1 and L929 cells and BALB/C mice were purchased from the Pasteur Institute (Tehran, Iran) and Royan Institute (Isfahan, Iran), respectively.

### 2.2. Preparations

Synthesis of (NH_4_)_3_{[Gd(Hina)_2_(ina)(H_2_O)_2_][MnMo_9_O_32_]}_2_·12H_2_O, (GdMnMo; POM)

Step A: (NH4)6(Mo7O24)·4H2O (2.002 g, 1.620 mmol) was dissolved in 20 mL of distilled water under stirring, and the pH value was adjusted with glacial acetic acid at the limit of 5. After 15 min, Mn (CH_3_COO)_2_·4H_2_O (0.120 g, 0.490 mmol) and 0.45 mL of 30% H2O2 solution were added and stirred for a further 15 min. Then the resulting mixture was placed in a water bath at 90 °C for 1 h, and the precipitates were filtered quickly. By cooling the filtrate, orange-red crystals were formed, and then 15 mL of distilled water was added and heated for 5–10 min until the crystals dissolved [22].

Step B: Gd(NO_3_)_3_·6H_2_O (0.250 g, 0.577 mmol and Hina (0.269 g, 2.185 mmol) were dissolved in 10 mL of distilled water, 2 mL of GAA, and 5 mL of DMF added under stirring; the mixture was stirred for 30 min, at room temperature. Finally, solution B was added to solution A and stirred for 20 min, at room temperature, and then kept for 2 h in a water bath at 60 °C. It was then filtered, and the solution was evaporated slowly at room temperature to obtain the product crystals; after two days, orange crystals were obtained [22]. Then different characterization techniques, including Fourier transform infrared (FTIR), Inductively coupled plasma optical emission spectrometry (ICP-OES), CHNS, UV–visible, X-ray diffraction (XRD), vibrating sample magnetometer (VSM), dynamic light scattering (DLS), and Zeta potential, were performed on the synthesized compounds.

#### 2.2.1. Synthesis of Chitosan–Imidazolium Conjugate (CSIm)

N-methylimidazole (10 mL, 0.16 mol) was added dropwise and very slowly to a mixture of 1,2-dichloromethane (25 mL, 0.32 mol) in acetone (25 mL), at room temperature. The resulting mixture was stirred for 48 h at 75 °C. When the reaction was completed, the product precipitates were obtained by adding ethanol to the reaction mixture. In the next step, it was washed with anhydrous ethanol and dried under vacuum at 80 °C. The chemical structure of this conjugate was confirmed by FTIR, XRD, and ^1^HNMR spectra [20,23].

#### 2.2.2. Preparation of POM@CSIm NPs

A total of 40 mg of chitosan–imidazolium polymer was dissolved in 0.1% aqueous acetic acid solution (acetic acid (0.1%) aqueous solution was added drop by drop until the CSIm (shell) dissolved; we tried to maintain the pH below 5), and 40 mg of POM was added to the above solution and stirred for 8 h, at room temperature, and then using a 12 kDa impurity dialysis membrane. Unloaded items were removed. The mixture inside the dialysis bag was lyophilized powder by a freeze dryer (Christ Alpha 1–4) machine, with an added cryoprotectant [21,24]. Moreover, for the physicochemical characterization of POM@CSIm FTIR, DLS, Zeta potential, scanning electron microscope (SEM), VSM, and ICP-OES were performed.

### 2.3. Characterizations

FTIR spectra (KBr pellet) were recorded on an FTIR (6300, Jasco, Tokyo, Japan) instrument in the range of 350–7800 cm^−1^. The mean particle size distributions and surface charges of NPs were determined using the DLS and Zeta potential (Zetasizer-ZEN 3600 Malvern Ltd., Worcestershire, UK). The surface morphology of the NPs was studied by SEM (LEO 1430VP), and flow cytometry studies were carried out on BD FACS-Calibur flow cytometer (Becton Dickinson, Franklin Lakes, NJ, USA). Images were taken using visible or fluorescent light, using a fluorescent microscope (V-670, Jasco, Tokyo, Japan). ICP-OES was carried out on an ICP-OES (Varian Vista-Pro, Ottoway, Australia). X-ray diffraction analysis was performed using XRD (Bruker Co., Bremem, Germany). A CHNS analysis was carried out on (Costech, Milan, Italy) and VSM (MDKB, Meghnatis Daghigh Kavir Yazd, Kashan, Iran). ^1^H NMR spectra were recorded on an NMR (Bruker Biospin AC-80, 400 MHz, Ettlingen, Germany) spectrometer, with deuterated acetic acid, water, and DMSO as solvents, at 25 °C, and chemical shifts were recorded in ppm relative to tetramethylsilane (TMS). Ultraviolet spectra were recorded using (Shimadzu UV-160, Kyoto, Japan) a UV–visible spectrophotometer.

### 2.4. Relaxivity Measurements

T_1_-weighted MR imaging of the nanocluster was performed by the 1.5 Tesla clinical MRI system (Magnetom Aera, Siemens, Hanover, Germany), using a 32-channel head coil. For this method, samples were dispersed in distilled water at different concentrations (0, 0.12, 0.24, 0.36, 0.48, 0.60, 0.72, and 0.84 mM), and all samples were dispersed in 2% agarose solution in a series of microtubules and prepared for MR imaging. All T_1_-weighted images were obtained using multi-spin echo pulse sequence by the following parameters: echo time (T_E_) = 9 ms; repetition time (T_R_) = 100, 200, 300, 500, 700, 1000, and 1500 ms; field of view (FOV) = 87.5 × 200 mm^2^; slice thickness = 3 mm; flip angle = 90°; and voxel size = 0.8 × 0.8 × 3 mm. MR imaging data were analyzed using an in-house script developed in R2022a (MATLAB 9.12) software. In the T_1_ mapping method, one-parameter models were implemented, and a region of interest (ROI) was semi-automatically plotted by using the Hough Transform technique for edge detection and segmentation in MATLAB software. Then T_1_ values were calculated in terms of the voxel in the ROI selected by the equation below, and Levenberg–Marquardt nonlinear least-squares regression was performed on the dataset.
$$\mathrm{SI}\;\left(\mathrm{voxel}\;\left(i,j\right)\right)\;=S_{0}\;\left(\mathrm{voxel}\;\left(i,j\right)\right)\;[1-e{(}^{\frac{-\mathrm{TR}}{T1}}\left(\mathrm{voxel}\;\left(i,j\right)\right)]$$
where SI is the signal intensity of each concentration of the samples, and S_0_ is the signal intensity of water. Then the mean and standard deviation measures for all voxels within the ROIs were computed. The R_1_ relaxation rate was then plotted as a function of different concentrations by the following equation:R_1_ = 1/T_1_

Finally, the longitudinal relaxivity (r_1_) was calculated by the linear least-squares regression on this plot and the goodness of fit for evaluation, using the adjusted R_1_ metric.

### 2.5. Ex Vivo Hemolysis Assessment

Whole-blood samples were prepared from the Isfahan blood bank in heparin tubes to investigate the biocompatibility of POM@CSIm nanoparticles. Then, to separate the red blood cells (RBCs), the blood sample was centrifuged at 3000 rpm for 5 min. In the next step, the obtained RBCs were washed with a sterile isotonic solution of 0.9% sodium chloride and suspended in the same solution with a ratio of 1:20. POM@CSIm nanoparticles were dispersed in 0.9% isotonic NaCl solution with a concentration of 300 μg/mL. Then 1 mL of colloidal solution was added to tubes containing 1 mL of red blood cell suspension. To prepare the control (complete hemolysis, 100%) and negative control (no hemolysis, 0%), 1 mL of RBC suspension was mixed with 1 mL of deionized water and 0.9% NaCl isotonic solution, respectively. After 4 incubations at 37 ± 1 °C, with shaking, in a water bath, the tubes were centrifuged at 3000 rpm for 5 min, and the absorbance of the supernatant was determined by UV–visible spectroscopy at a 540 wavelength [20]. Finally, the hemolysis percentage of the samples was calculated using the following equation:$$\mathrm{Hemolysis}\;(\%)=\;\frac{A_{s}-A_{\mathrm{NC}}}{A_{\mathrm{PC}}-A_{\mathrm{NC}}}\times 100$$
where A_s_ refers to the absorption of sample, A_NC_ means the absorbance of negative control, and A_PC_ means absorption of positive control.

### 2.6. Cell-Based In Vitro Studies

#### 2.6.1. Cell Culture

In this study, breast cell carcinoma (4T1) and normal mouse skin cells (L929) were used. The cells are purchased from the cell bank of Pasteur Institute, Tehran, Iran. Cells in 75 mL flasks with RPMI 1640 culture medium for both cell lines containing 10% fetal bovine serum and 1% pen/strep antibiotics in a wet incubator at 5% carbon dioxide at 37 °C were cultured and propagated. Both cancerous and normal cells used in this study are sticky and grow as a single layer in the culture medium and adhere to the bottom of the flask. For each test, cells are detached from the plate by using trypsin and counted and utilized by using a hemocytometer. The cell culture process was performed according to standard cell culture protocols, including cell de-freezing, cell counting, cell passage, and cell freezing under controlled conditions, such as sterile culture medium, suitable pH, and temperature–humidity stability.

#### 2.6.2. MTT Assay

To assess the toxicity and cell biocompatibility of the nanocluster, an MTT test was used. The 4T1 and L929 (1 × 10^4^ cells/well) cells were seeded in 96-well plates and were incubated at 37 °C under a humidified 5% CO_2_ atmosphere for 24 h. Then POM (core), manganese acetate, POM without gadolinium (A), and POM@Cs-Im (core/shell) with different concentrations were added to the cells and incubated for another 24 h. The medium was removed, and 100 μL of RPMI 1640 containing MTT (5 mg/mL) solution was added to the wells; the plates were then kept in the dark conditions for 4 h. The supernatant was then removed, 100 μL of DMSO was added to each well, and the cells were incubated by shaking for 10 min to dissolve the formazan crystals. Finally, the optical density (OD) of each sample was measured using an ELISA (Bio-Rad, Hercules, CA, USA) at 570 nm. Cell viability was calculated as a percentage with the ratio of OD values in the treatment and control groups. Each experiment was repeated three times under the same conditions.

#### 2.6.3. Cellular Uptake

A fluorescence microscope and flow cytometer were used for the uptake cellular. For this purpose, 4T1 and L929 were seeded in 6-well plates (3 × 10^6^ cells per well) and were incubated at 37 °C under a 5% CO_2_ atmosphere for 24 h. After that, the cells were treated with a nanocluster containing curcumin (curcumin was added into the POM@CsIm NPs during the preparation step), with 100 and 500 μg/mL, and incubated for 24 h. Then the supernatant was removed, the cells were washed three times with PBS, and they were harvested by trypsin and counted. Subsequently, they were centrifuged and assessed with a fluorescence microscope (Nikon Company, Melville, NY, USA) and flow cytometer (BD FACSCalibur, London, UK). All measurements were performed in triplicate.

### 2.7. In Vitro MR Imaging

T1-weighted images on 4T1 and L929 were performed using clinical MR imaging system, as mentioned before. For these experiments, cells were seeded in 6-well plates (3 × 106 cells per well) and maintained for 24 h in 5% CO_2_ at 37 °C. The cells were incubated with polyoxometalate gadolinium–manganese at different concentrations for 6 h. They were washed three times with PBS, separated using trypsin, and suspended in 2% agarose gel (KBC). Finally, T1-weight MR images were recorded using the following parameters: flip angle = 90°, TE = 8.9 ms, TR = 400, voxel size = 0.6 × 0.6 × 3 mm, slice thickness = 3 mm, and FOV = 192 × 220 mm^2^.

### 2.8. In Vivo Studies

#### 2.8.1. Animal Caring

Animal care before and during the experimental procedures was performed based on the European Community guidelines and was approved by the local ethical committee, Isfahan University of Medical Sciences (IUMS), Isfahan, Iran (Approval Number: IR.MUI.MED.REC.1400.458). All animals were housed under standard laboratory conditions.

#### 2.8.2. Tumor Induction

To induce tumor in mice, approximately 1 × 106 4T1 cells were injected subcutaneously into the lateral region of the mouse. In this study, six groups of BALB/C mice were considered, including the control group and group therapy at different times (0 (before injection), 2, 4, 6, and 24 h after injection) [25]. MR imaging of mice was performed before and after injection of nanocluster. Then, using ImageJ software and drawing the region of interest (ROI) in the images, the signal intensity before and after the injection of the nanocluster was determined and checked.

#### 2.8.3. In Vivo MR Imaging

For in vivo MR imaging, 4-to-6-week-old female mice bearing 4T1 tumors were anesthetized by injecting 10 and 100 mg/kg of xylazine and ketamine, respectively [26]. Then a 2 mg/kg nanocluster was intravenously (i.v.) injected into the tail vein of the mice, and T_1_-weighted MR images were acquired by the same MR scanner, as introduced in Section 2.7, using the head coil. MR imaging protocol was performed using the following parameters: T_E_ = 8.9 ms, T_R_ = 500 ms, slice thickness = 3 mm, matrix size = 192 × 154, FOV = 110 mm^2^, flip angle = 90°, and voxel size = 0.7 × 0.7 × 3 mm. The MRI average signal intensity of tumors at each time was calculated by drawing an ROI on the tumor region, using MATLAB software.

All raw data were processed and quantified using the R2022a (MATLAB 9.12) software, and statistical evaluations were applied using the IBM SPSS Statistics software (version 20.0.0, IBM Corp, Armonk, NY, USA).

### 2.9. Statistical Analysis

Data were analyzed using SPSS software (version 20.0.0) and were demonstrated as mean values ± SD of experiments. Statistical analyses were performed using the two independent samples *t*-test method, and *p*-values < 0.05 were considered statistically significant.

## 3. Results and Discussion

### 3.1. Characterization of NPs

In this study, (NH_4_)_8_{[Gd(Hina)(ina)(H_2_O)_2_][Mn^IV^Mo_9_O_32_]}_2_·12H_2_O POM@CSIm was successfully synthesized. In this structure, [MnMo_9_O_32_]^6−^ is a Waugh-type that is composed of a central MnO_6_ octahedron surrounded by three MoO_6_ octahedrons (Mo_1_, Mo_5_, and Mo_9_) arranged on the three vertices of a triangle, giving rise to an Mn-centered triangular {MnMo_3_O_18_} cluster. Then two edge-sharing Mo_3_O_13_ triads (one Mo_3_O_13_ triad is formed by Mo_2_, Mo_3_, and Mo_4_, and the other Mo_3_O_13_ triad is constructed from Mo_6_, Mo_7_, and Mo_8_) are, respectively, capped above and below the Mn-center triangular {MnMo_3_O_18_} cluster. Generally, organic ligands such as Hina can effectively coordinate with Gd^3+^ ions to form Gd^3+^ organic complexes, preventing sediments from directly combining POM precursors and Gd^3+^ ions. Based on the result obtained for solution A, polyoxoanion [MnMo_9_O_32_]^6−^ will not be formed if the hydrogen peroxide used is too much or too little. It has been found that the optimal use of hydrogen peroxide (30%) is 0.45–0.9 mL. Moreover, for solution B, the optimal amount of Gd^3+^ (NO_3_)_3_·6H_2_O is in the range of 0.20–0.27 g. If the value of Gd^3+^ (NO_3_)_3_·6H_2_O exceeds 0.27 g, precipitation is formed, and if it is less than 0.20 g, the product yield is very low, or the target product is not even obtained [22].

For the physicochemical characterization of POM, FTIR (Appendix A), the elemental analysis included an ICP-OES and CHNS analysis, and the UV–visible (Appendix A), VSM, DLS and Zeta potential (Appendix A) were performed. FTIR (Appendix A), ^1^HNMR (Appendix A), and XRD (Appendix A) for the physicochemical characterization of the shell were performed; and for the physicochemical characterization of POM@CSIm, FTIR (Appendix A), DLS, Zeta potential (Appendix A), SEM, VSM, and ICP-OES were performed, as shown in Figure 1, Figure 2, Figure 3 and Figure 4 and Table 1 and Table 2.

According to Figure 1A, the absorption band center at 898 and 940 cm^−1^, with a little displacement, corresponds to ν (Mo–O_t_); those at 678 cm^−1^ and 593 cm^−1^ are assigned to the ν (Mo–O_b_–Mo) stretching vibration, and the absorption band appearing at 540 cm^−1^ is attributed to the Mn−O stretching vibration. The strong broad band at around 3200–3500 cm^−1^ is assigned to the stretching vibration mode of coordinate and lattice water molecules. The carboxylic group is anticipated to show very intense absorption bands from the asymmetric (1500–1630 cm^−1^) and symmetric (1350–1460 cm^−1^) stretching vibration. The absorption bands observed at 1604–1406 cm^−1^ are attributed to the asymmetric [νas (CO_2_^−^)] and symmetric [νsy (CO_2_^−^)] stretching vibrations of the carboxylic groups of the ligands. The signal appearing at 3104 cm^−1^ is attributable to the ν (N−H) stretching vibration of NH_4_. All of these proofs were in good agreement with the previous report [22].

According to Figure 1B, the stretching vibrations of the imidazolium ring, which can be seen in the absorption bands in the region of 1657 and around 3078 cm^−1^, confirm the conjugated structure of CSIm [27]. Moreover, the stretching bands around 3432 correspond to N-H and O-H bonds, and the absorptions in 1657 and 1048 are characteristic features of the C=O and N-H bending models.

According to Figure 1C, the absorption peaks related to the POM and CSIm can confirm the coating of the structure.

For the elemental analysis for C_24_ H_82_ N_12_ O_88_ Gd_2_ Mn_2_ MO_18_, using ICP-OES and CHNS for the metal contents of POM, the theoretical percent values are as follows: Gd: 7.67; Mn: 2.68; and Mo: 42.14. The values found by ICP-OES are Gd:7.76; Mn:2.78; and Mo:42.29. Theoretical values: C, 7.03; H, 2.02; and N, 4.10. Values found by CHNS analysis: C, 7.23; H, 1.87; and N, 3.94.

The UV–visible absorption spectrum absorption band (Figure 2) at 340 nm is attributed to the characteristic excitation of Mn^4+^ ions that confirms the presence of Mn in a higher oxidation state in the POM [28]. The absorption spectrum absorption band at 478 nm is attributed to the ^4^T_2g_ → ^4^A_2g_ transitions of the Mn^4+^ center in the [MnMo_9_O_32_]^6−^ polyoxoanion [22].

According to Figure 3, POM magnetization showed enlarged view around −140 to 140 Oe (magnetization −0.25–0.25) applied magnetic field. Comparing the VSM of POM with the VSM of the POM@ CSIm, it is clear that the symmetry is preserved, but the magnetic property of the nanocluster decreased after coating it with CSIm (−0.1–0.1). The reason for this reduction can be the reduction of saturation magnetization per unit mass caused by the reduction of the magnetic moment at the POM@CSIm interface due to the interaction of the POM with the Cs-Im and the shielding effect induced by the CSIm layer around the surface of the POM [29].

From comparing the Zeta potential of the POM with the Zeta potential of the POM@CSIm, it is clear that the value of the Zeta potential is improved, which can be the result of the successful conjugation of the POM with the CSIm [13].

According to the HNMR, the shell signals around 7.5 and 8.7 ppm proved the presence of the imidazolium aromatic features of CSIm. That is in agreement with the previous study [20].

Scanning Electron Microscopy (SEM) images were prepared to check the shape and size of POM@CSIm nanoclusters. As shown in Table 1, the nanoclusters have an average size that is lower than DLS (393.1 nm) because DLS is taken in the solution phase and gives the hydrodynamic size, but SEM is obtained in the solid phase and shows a smaller size compared to DLS.

Based on the ICP-OES results, according to the weight ratio of gadolinium in the formula of the POM molecule, the percentage of POM loading in the nanocluster can be calculated using the following equation:$$loading\;content\;percent\left(\%\right)=\;\frac{\left(\frac{The\;value\;of\;gadolinium\;obtained\;in\;ICP-OES}{10}\right)\times \left(\frac{MW\;of\;POM}{total\;weight\;of\;Gd\;in\;POM\;(exact\;mass)}\right)}{weight\;of\;NPs\;(for\;ICP-OES\;exam)}\times 100$$
$$loading\;content\;percent\left(\%\right)=\frac{\left(\frac{42.6}{10}\right)\times \left(\frac{4134.25}{315.85}\right)}{10}\times 100=58.3$$

### 3.2. Relaxivity Measurements

To evaluate the potential application of POM@CSIm nanoclusters as an MR nano-agent, their magnetic relaxation properties were investigated. The results showed that the nanoclusters effectively shortened T_1_ and significantly increased the signal intensity at T_1_. According to the slope of the curve, the r_1_, corresponding to POM@CSIm, was obtained as 4.71 mM^−1^ s^−1^ (Figure 5), which is higher than that of Magnevist (r_1_ = 3.80 mM^−1^ s^−1^) [30] and Dotarem (2.79 mM^−1^ s^−1^) [31].

### 3.3. In Vitro Cell Based Assessments

#### 3.3.1. MTT Assay

The L929 and 4T1 cells were used to evaluate the cytotoxicity of the compounds used to construct nanoclusters. As shown in Figure 6, L929 does not indicate a significant decrease in viability for 24 h in the four compounds studied and remains almost above 75%. Over the course of 48 h, the viability decreases, especially with an increasing concentration. The comparison between the cytotoxicity of three groups (manganese acetate, A; MnMo_6_O_16_; and POM-core) in the L929 cell line shows that there is no significant difference between the groups (*p* = 0.4614). In 4T1 cells, the 24 h time does not significantly reduce viability. It remains above 50%, which can be concluded that the cytotoxicity of the compounds in cancer cells is higher than in L929 cells. Moreover, a comparison between the cytotoxicity of three groups (manganese acetate, A; MnMo_6_O_16_; and POM-core) in the 4T1 cells shows that there is a significant difference between the groups (*p* = 0.0001). According to the comparison between the cytotoxicity of the core and core/shell in both cell lines, the shell has reduced toxicity. In addition, a comparison between the cytotoxicity of core/shell in the 4T1 and L929 cell lines shows that there is a significant difference between the groups (*p* = 0.00169). In general, based on the graphs obtained in the four investigated compounds, it is clear that the toxicity is higher after 48 h than after 24 h.

#### 3.3.2. Cellular Uptake

A qualitative evaluation of the nanocluster was performed by 4T1 and L929 cells by fluorescence imaging and flow cytometry. Figure 7A show the cellular uptake obtained by fluorescence microscopy images of L929 and 4T1 cells after 24 h of incubation at a concentration of 500 μg/mL, respectively. The results of fluorescence imaging showed that nanoclusters in 4T1 cells had more cellular uptake than L929 cells, and a weak fluorescence signal was observed in L929 cells. Moreover, the results of the flow cytometry (Figure 7B) of L929 and 4T1 cells after 24 h of incubation at 100 and 500 μg/mL concentrations showed that all cells in the control group are located in the M1 region, and this region is defined in such a way that there is no cell in the M2 area. The movement of the graphs to the right, i.e., from M1 to M2 in the FL-1 channel, indicates the number of nanoclusters entering the cells. The cellular uptake of nanocluster by 4T1 cells was 44.2 and 78.3 at 100 and 500 μg/mL concentrations, respectively. Meanwhile, as shown in the image, L929 cells showed a small uptake of nanoclusters, with the uptake of 11.9 and 22.3 at concentrations of 100 and 500 μg/mL, respectively. This difference is not statistically significant (*p* > 0.05). Moreover, based on the results, the cellular uptake of nanoclusters by 4T1 cells was much higher than for L929 cells (*p* < 0.05).

### 3.4. Ex Vivo Hemolysis Assay

The RBC hemolysis assay is a valid test to measure the safety of any biomaterial in ex vivo conditions [32]. The nanoparticles may rupture the RBC membrane and release hemoglobin, which can be measured by UV–vis spectrometry. Based on the obtained results, POM@CSIm NPs induced RBC hemolysis below 4%, suggesting POM@CSIm NPs for a wide safety margin in blood-contact applications and suitability for intravenous administration. Mahvash et al. [20] reported the same results for Anderson-type manganese polyoxomolybdate entrapped in a CSIm shell.

### 3.5. In Vitro Imaging

The MR images (Figure 8A,B) show the in vitro image of L929 and 4T1 cells that shows an increase in the signal intensity in 4T1 compared to L929 cells and shows increased intensity with an increased concentration. Moreover, a comparison between 4T1 and L929 cells (see Figure 8C) showed that the signal intensity increases with the increase in the nanocluster concentration and showed more signal intensity in 4T1 cells (125.66 ± 21.57) compared to L929 cells (93.67 ± 39.11) (*p* < 0.05).

### 3.6. In Vivo Imaging

The T_1_-weight MR images (Figure 9) show the signal intensity after the injection of nanoclusters in the tumor area of BALB/C. As shown in this figure, 4 h after the nanoclusters injection, there is the highest accumulation of nanoclusters in the tumor area and, as a result, the highest signal intensity, and after this time, the signal intensity decreases. Moreover, the comparison of tumor-bearing mice with control mice (without tumor and injection) in Figure 10A,B shows that the signal intensity is higher in tumor-bearing mice (*p* < 0.05), and the passage of time does not change the intensity of the signal in the control mice.

## 4. Conclusions

In recent years, the use of nanoparticles as contrast agents in MR imaging for the early detection of cancers has been considered. In this study, nanoclusters with a chitosan–imidazolium coating were successfully synthesized and characterized. In vitro toxicity studies showed that nanoclusters have excellent biocompatibility, and chitosan–imidazolium as a shell can significantly improve biosafety and reduce the toxicity risk of polyoxometalate as an MRI contrast agent. Moreover, the MRI results showed selective accumulation and enhanced intracellular uptake POM@CSIm nanoclusters in 4T1 compared to L929 cells. The in vivo results showed that nanoclusters can passively target and selectively accumulate in the tumor area especially 4 h after injection. Overall, with the satisfactory of the results, high compatibility, and good tumor uptake, POM@CSIm has considerable promise as an MRI nano-agent for early 4T1 cancer detection.

## Figures and Tables

**Figure 1 micromachines-14-00741-f001:**
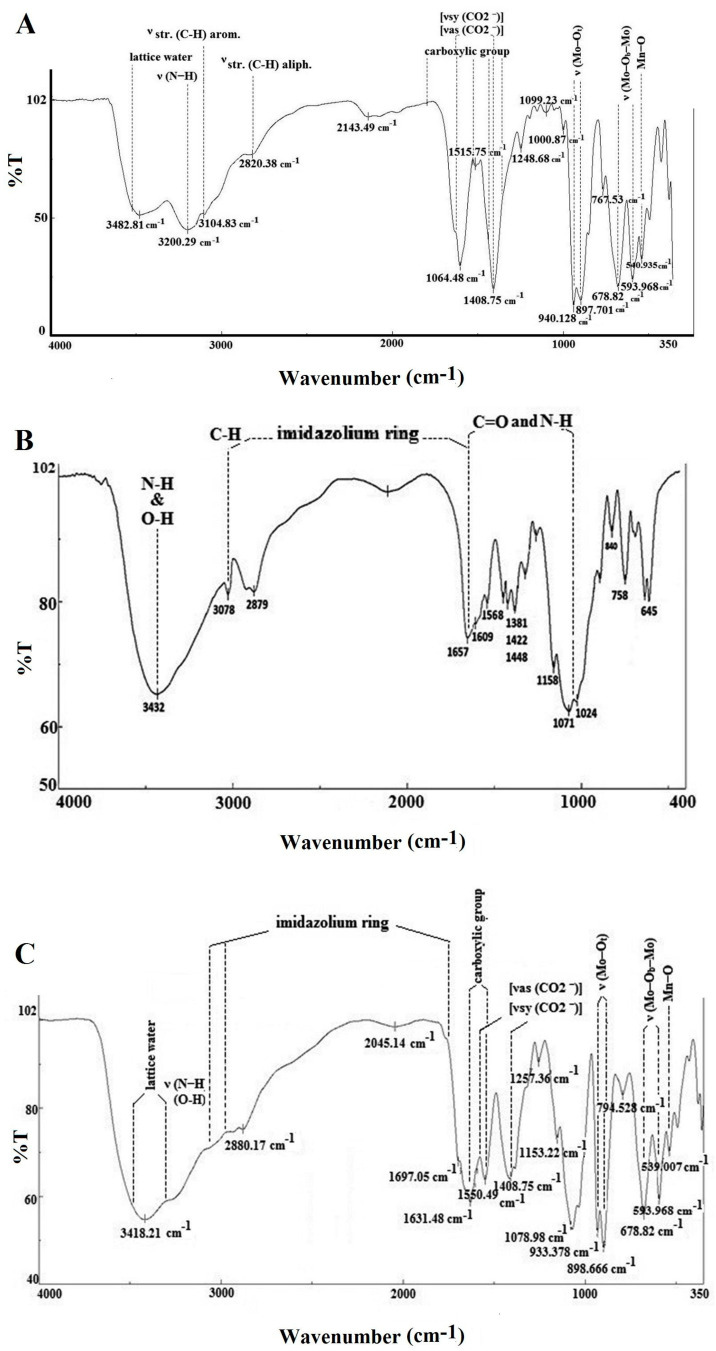
(**A**) FTIR results of POM, (**B**) FTIR of CSIm shell, and (**C**) FTIR of POM@CSIm.

**Figure 2 micromachines-14-00741-f002:**
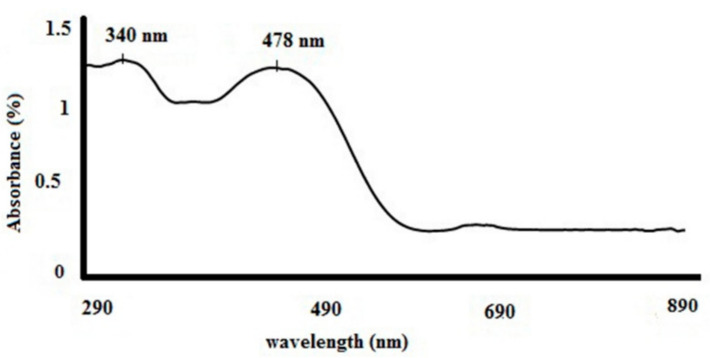
The UV–visible absorption spectrum results of POM in the wavelength range 190–900 nm.

**Figure 3 micromachines-14-00741-f003:**
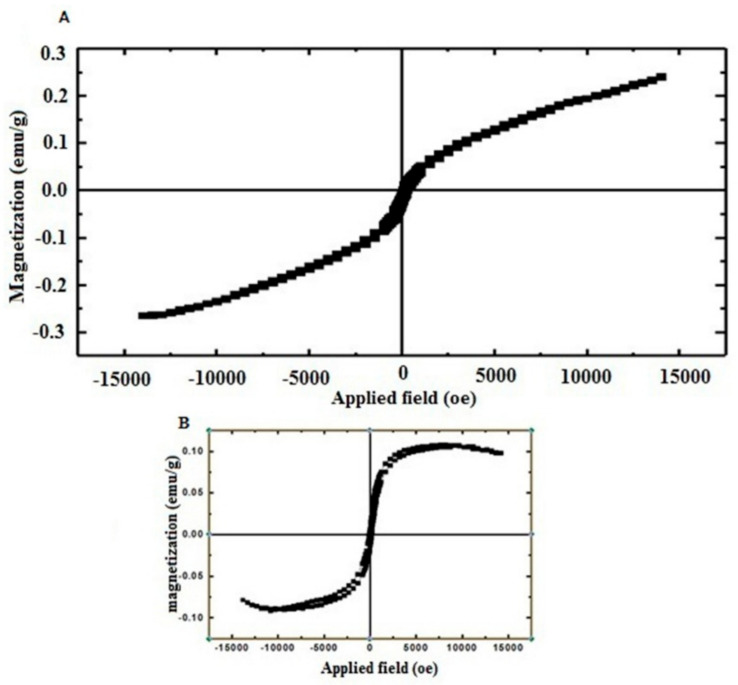
Vibrating sample magnetometer (VSM) results of (**A**) POM and (**B**) POM@CSIm.

**Figure 4 micromachines-14-00741-f004:**
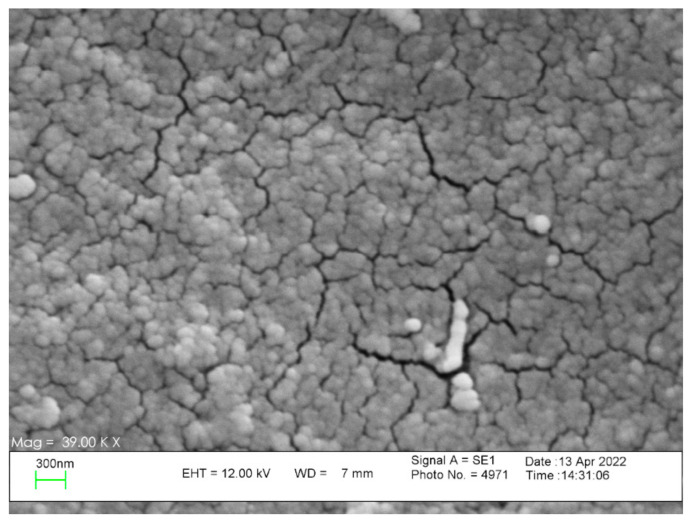
SEM results of POM@CSIm NPs.

**Figure 5 micromachines-14-00741-f005:**
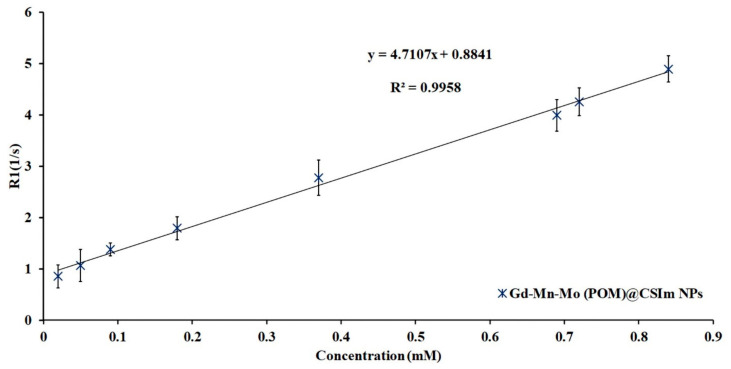
Relaxivity (r_1_) plots of POM@CSIm at various concentrations of nanocluster.

**Figure 6 micromachines-14-00741-f006:**
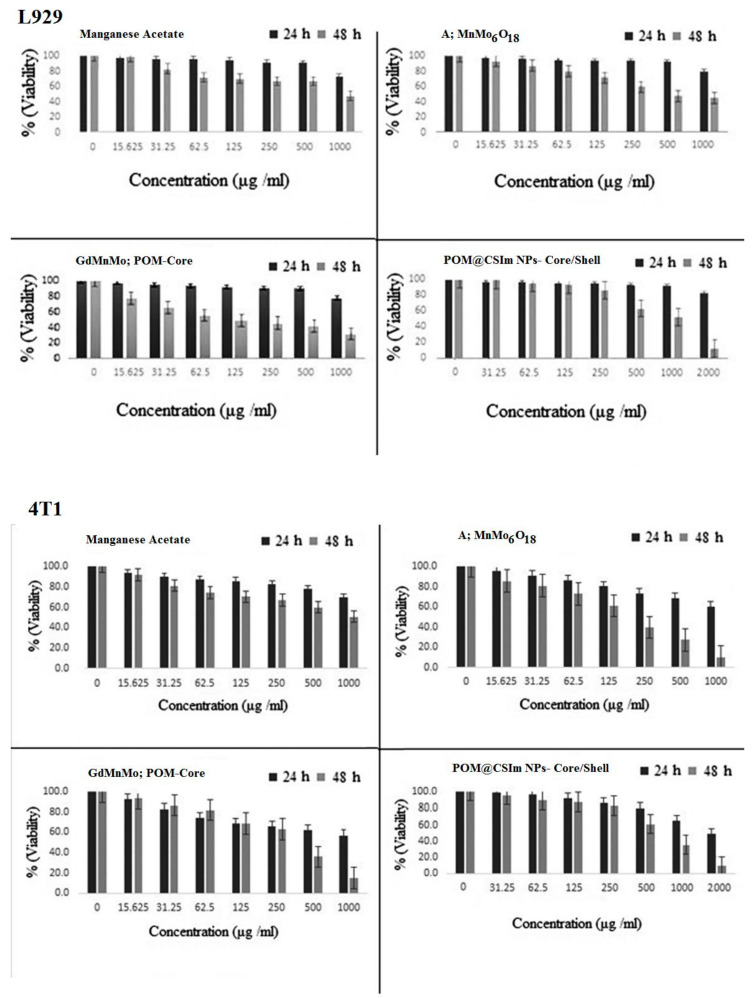
Cytotoxicity results of (GdMnMo) POM@CSIm on L929 and 4T1 for four compounds: manganese acetate 24 h and 48 h, A; MnMo_6_O_18_ 24 h and 48 h; POM-core 24 h and 48 h; and core/shell 24 h and 48 h.

**Figure 7 micromachines-14-00741-f007:**
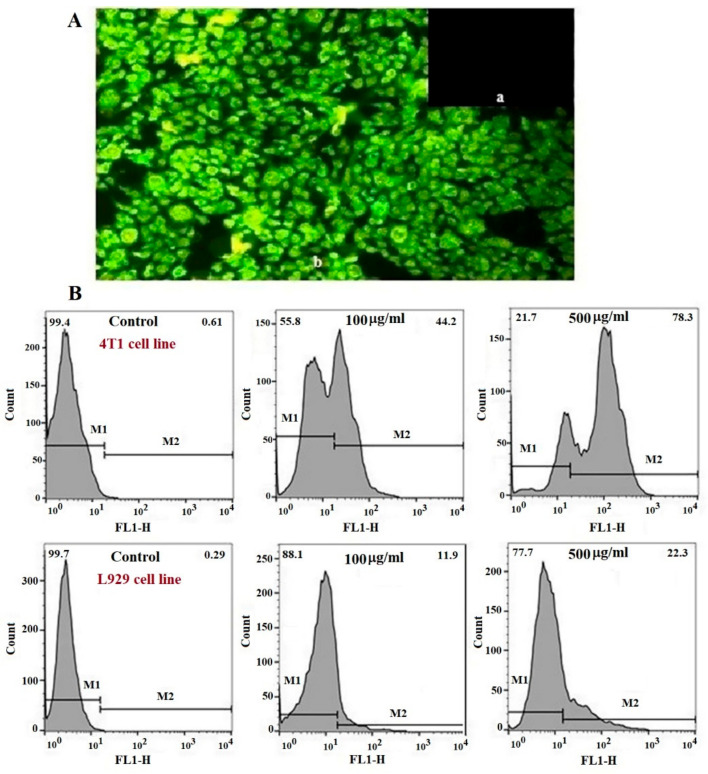
(**A**): (**a**) Cellular uptake of L929 by fluorescence microscopy in 500 μg/mL after 24 h. (**b**) Cellular uptake of 4T1 by fluorescence microscopy in 500 μg/mL after 24 h. (**B**) Cellular uptake of control, L929, and 4T1 by flow cytometry in 100 and 500 μg/mL after 24 h.

**Figure 8 micromachines-14-00741-f008:**
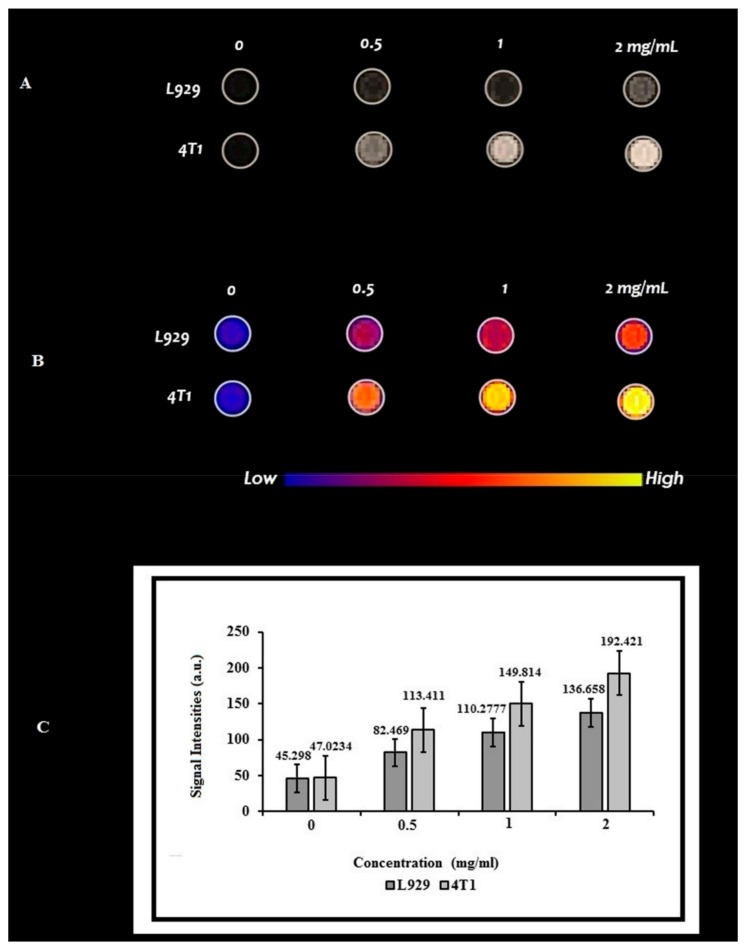
(**A**) MRI grayscale image of L929 and 4T1 at 0, 0.5, 1, and 2 mg/mL. (**B**) Color MR image of L929 and 4T1 at 0, 0.5, 1, and 2 mg/mL. (**C**) Comparison of signal intensity between 4T1 and L929 cells.

**Figure 9 micromachines-14-00741-f009:**
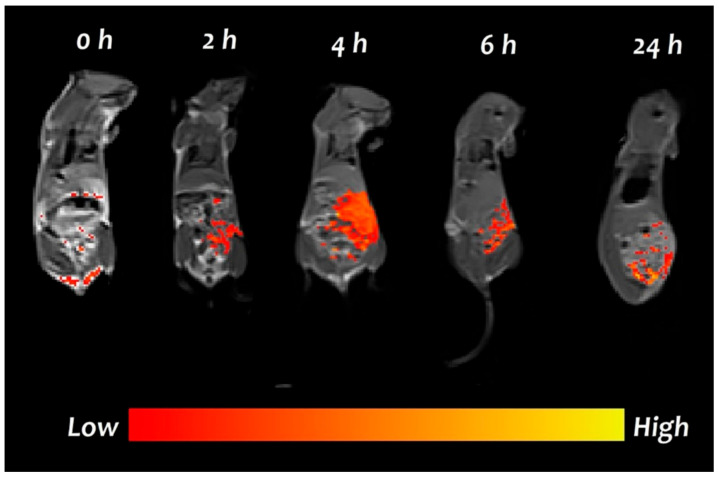
MR images of BALB/C mice after injection (0 h, 2 h, 4 h, 6 h, and 24 h) with highest accumulation of nanoclusters in the tumor area after 4 h.

**Figure 10 micromachines-14-00741-f010:**
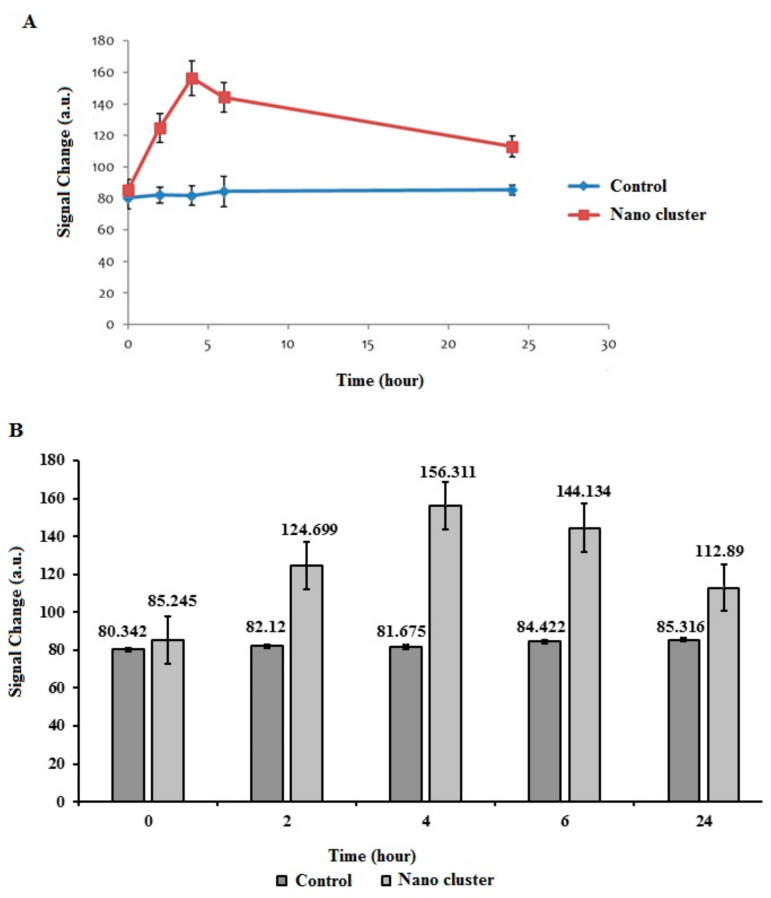
Both (**A**,**B**) show signal intensity changes over time in tumor-bearing and control mice.

**Table 1 micromachines-14-00741-t001:** Mean DLS and Zeta potential and electrophoretic mobility.

Sample	Hydrodynamic Size (nm)	Zeta Potential (mV)
NPs	250.5 nm ± 17.9 nm	−35.5 mV
POM@CSIm NPs	393.1 nm ± 22.4 nm	−30.6 mV

**Table 2 micromachines-14-00741-t002:** ICP-OES results of POM@CSIm.

Equipment	Element	Value (mg/L)
ICP-OES	Gd	42.68
ICP-OES	Mo	232.60
ICP-OES	Mn	15.29

## Data Availability

The data presented in this study are available upon request from the corresponding author.

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
