# Peer review of "Chitosan–Imidazolium Core–Shell Nanoparticles of Gd-Mn-Mo Polyoxometalate as Novel Potential MRI Nano-Agent for Breast Cancer Detection"

_micromachines, 2023, doi:10.3390/mi14040741_

Round 1

Reviewer 1 Report

I accurately reviewed the manuscript from Aminolroayaei et al. concerning the synthesis, characterization and bio evaluation of Chitosan-Imidazolium Core-Shell Nanoparticles of Gd-Mn-Mo Polyoxometalate.

The overall manuscript appears to be well written with clear goals and means. I have a few comments nonetheless : 

- Authors should pay attention accronyms as some are not defined or defined after the first appearence. 

- In paragraph 2.2 (and the same comment can apply elsewhere in the experimental section), volumes are not specified : "a large amount of aqueous ammonia" for example. Authors should also add the concentrations of the used solutions. The same comment apply line 156 : "dissolved in 0.1 % aqueous..." 

- Figure 1 to 10 are used to describe the particles characterization but in my opinion some of them could be merge to reduce the number of displayed figures. As an example, FTIR figures could be presented under the same figure. Also, some of the figure quality could be improve as some of them appeared blury. Figure 8 is barely used in the text and could be deleted as it only confirm result obtained in previous study.

- Figure 10 is quite unclear in my opinion. I don't understand how the authors calculate/measure the size of their particule as no value/explanation was given. Also it might be a mistake line 395 : "... size lower than DLS (3093.1 nm) ...". 

-I don't understand the equation used to calculate POM loading. The equation involved the mass of particles used. But the particles were obtained after liquid synthesis and evaporation. Have the authors check the remaining solvent before weighting their sample ?

- In figure 11 : R² could be included with the linear equation.

-Finally, in my opinion the term "nanoparticle" is inapropriate as the nanosize range run from 2 to 100 nm in one of the 3 dimensions. 

Overall, once these minor comments/questions are resolved, the manuscript could be published.

Best regards.

Author Response

Reviewer 1:

Thanks to your useful and valuable points. I did correct your comments in the text and highlighted them with yellow color.

The overall manuscript appears to be well written with clear goals and means. I have a few comments nonetheless: 

- Authors should pay attention accronyms as some are not defined or defined after the first appearence.

It was corrected.

- In paragraph 2.2 (and the same comment can apply elsewhere in the experimental section), volumes are not specified: "a large amount of aqueous ammonia" for example. Authors should also add the concentrations of the used solutions. The same comment apply line 156: "dissolved in 0.1 % aqueous..." 

The method sections regarding synthesis of POM and POM@CSIm have been modified. Please check revised version of the text on page 3, as well as page 4.

- Figure 1 to 10 are used to describe the particles characterization but in my opinion some of them could be merge to reduce the number of displayed figures. As an example, FTIR figures could be presented under the same figure. Also, some of the figure quality could be improve as some of them appeared blury. Figure 8 is barely used in the text and could be deleted as it only confirm result obtained in previous study.

It was corrected.

- Figure 10 is quite unclear in my opinion. I don't understand how the authors calculate/measure the size of their particule as no value/explanation was given. Also it might be a mistake line 395: "... size lower than DLS (3093.1 nm) ...” 

The numbers are reported through DLS and we provided the image to confirm the homogeneity of the nanoparticles and DLS numerical information.

It was corrected.

-I don't understand the equation used to calculate POM loading. The equation involved the mass of particles used. But the particles were obtained after liquid synthesis and evaporation. Have the authors check the remaining solvent before weighting their sample?

In successive vacuum drying cycles, we made sure that the sample was dry before measurement.

- In figure 11: R² could be included with the linear equation.

It was corrected.

-Finally, in my opinion the term "nanoparticle" is inapropriate as the nanosize range run from 2 to 100 nm in one of the 3 dimensions. 

The concept of nanoparticles in the applied field of medicine is different from its basic concept. In the field of medicine, larger sizes create an advantage for persistence in the biological system, one of which is accumulation in tumor tissue via passive targeting approach.

Overall, once these minor comments/questions are resolved, the manuscript could be published.

Reviewer 2 Report

The MS by Amin Farzadniya et al. describes new POM/chitosan composite for breast cancer detection. The presented research is a large work with a lot of methods and characterization. The research idea is clear and can be of interest for a wide readership. The MS is in the scope of Micromachines. So, the MS can be published in the journal after taking into account these points:

1) Characterization of POM. The powder diffraction pattern for the obtained crystalline material and theoretical data can not be compared. Please, do this analysis according to the standard procedure. Figure 3 is a bad joke at the current stage….

2) Fig. 4 has a strange capture. Vibrating sample magnetometer is a device but not a data…

3) Fig. 6. It is so clear why Zeta potential/electrophoretic mobility of POM and POM/chitosan particles are practically the same… This looks strange and should be clarified.

4) ICP-OES of Gd-Mn-Mo POM@Cs-Im demonstrates interesting results. If we compare these data with the same for pure POM it will give some strange points. According to the presented data Gd-Mn-Mo POM@Cs-Im contains a lot of Mn, but not POM. This should be clarified.

5) The quality of Fig. 14 should be improved.

6) Some additional experiments using “Mn2+” (e.g. as Mn-EDTA complex) as control should be done to estimate the effect of POM. It is clear that reported (https://doi.org/10.1039/C6CE02428A) POM is unstable in water and can be a source of other species.

Author Response

Response to the comments of Reviewer 2:

Thanks for your useful and valuable points. I did correct your comments in the text and highlighted them with yellow color.

The MS by Amin Farzadniya et al. describes new POM/chitosan composite for breast cancer detection. The presented research is a large work with a lot of methods and characterization. The research idea is clear and can be of interest for a wide readership. The MS is in the scope of Micromachines. So, the MS can be published in the journal after taking into account these points:

  • Characterization of POM. The powder diffraction pattern for the obtained crystalline material and theoretical data cannot be compared. Please, do this analysis according to the standard procedure. Figure 3 is a bad joke at the current stage….

We did not have a sample to compare, so we retrieved the XRD spectrum from the x-ray file related to the Gong P et al (2017) article, but in the final version, we only presented the XRD spectrum of our own sample, which you can see in the supplementary file.

2) Fig. 4 has a strange capture. Vibrating sample magnetometer is a device but not a data…

It was corrected.

3) Fig. 6. It is so clear why Zeta potential/electrophoretic mobility of POM and POM/chitosan particles are practically the same… This looks strange and should be clarified.

In the range of graphs provided by the laboratory, it seems that the two graphs are similar, but they have a significant difference in the peak, which is equivalent to the numbers provided.

4) ICP-OES of Gd-Mn-Mo POM@Cs-Im demonstrates interesting results. If we compare these data with the same for pure POM it will give some strange points. According to the presented data, Gd-Mn-Mo POM@Cs-Im contains a lot of Mn, but not POM. This should be clarified.

It was corrected.

5) The quality of Fig. 14 should be improved.

It was done.

6) Some additional experiments using “Mn2+” (e.g. as Mn-EDTA complex) as control should be done to estimate the effect of POM. It is clear that reported (https://doi.org/10.1039/C6CE02428A) POM is unstable in water and can be a source of other species.

For this reason, we used a polymer coating to protect it to increase its hydrolytic stability.

Reviewer 3 Report

Authors reported Chitosan-Imidazolium Core-Shell Nanoparticles of Gd-Mn-Mo Polyoxometalate as Novel Potential MRI Nano-agent for Breast 3 Cancer Detection work which is well characterized and showed good activity. 

Only need to revise the grammatical and typos errors to improv the manuscript.

Author Response

Response to the comments of Reviewer 3:

Thanks for your useful and valuable points. I did correct your comments in the text and highlighted them with yellow color.

Authors reported Chitosan-Imidazolium Core-Shell Nanoparticles of Gd-Mn-Mo Polyoxometalate as Novel Potential MRI Nano-agent for Breast 3 Cancer Detection work which is well characterized and showed good activity. 

Only need to revise the grammatical and typos errors to improve the manuscript.

All of the grammatical and typos errors were improved.

Round 2

Reviewer 2 Report

The MS has been improved and practically ready for publication. I have a few comments which should be corrected by the authors:

1) Fig. 5 is bad and should be changed.

2) Page 10. " The reason for this reduction can be the reduction of saturation magnetization per unit mass caused by the reduction of the magnetic moment at the POM@CSIm interface due to the interaction of the POM with the Cs-Im and the shielding effect induced by the CSIm layer around
the surface of the POM
"

This sentence should be modified.

3) Table 1 capture should be modified.

4) Table 2 is not mentioned in the main text near the table place.